# Textile Reinforced Concrete in Combination with Improved Self-Healing Ability Caused by Crystalline Admixture

**DOI:** 10.3390/ma13245787

**Published:** 2020-12-18

**Authors:** Hana Žáková, Jiří Pazderka, Pavel Reiterman

**Affiliations:** 1Faculty of Civil Engineering, Czech Technical University in Prague, Thákurova 7, 166 29 Prague, Czech Republic; jiri.pazderka@fsv.cvut.cz (J.P.); pavel.reiterman@fsv.cvut.cz (P.R.); 2University Centre of Energy Efficient Buildings, Czech Technical University in Prague, Třinecká 1024, 273 43 Buštěhrad, Czech Republic

**Keywords:** textile reinforced concrete, TRC, self-healing ability, crystalline admixture

## Abstract

The main aim of this study was to investigate the improved autogenous healing of concrete caused by a crystalline admixture in combination with textile reinforced concrete (TRC). This phenomenon (improved healing) has not yet been described by any independent study, and not at all in relation to TRC. The results of the study confirmed that the interaction between TRC and the crystalline admixture’s self-healing ability is advantageous and usable. The application of crystalline admixture could ensure the long-term entirety of the TCR element, where microcracks could occur. This allows for the creation of advantageous, thin (achieved by TRC) and waterproof (achieved by the crystalline admixtures) concrete structures. Moreover, this does not depend on temperature in the range of 4–30 °C (lower temperatures are of course problematic, as for most other cementitious materials). However, the interaction of both materials has its limits; the cracks must not be too wide (max. 0.1 mm), otherwise they will not heal. On the other hand, the advantage is that it does not matter what type of cement is used (CEM I and CEM II showed the same results), and the composition of the newly formed crystals in the cracks corresponds to the composition of the C-S-H gel, so it can be assumed that secondary hydration of the Portland cement occurred in the crack area.

## 1. Introduction

The problem of crack permeability in concrete is crucial for the assessment of any structure where there is a requirement for water impermeability. The concrete structures of buildings almost always have some cracks, due to uneven settlement or for other reasons. In the case where the substructure of the building is designed with waterproof concrete (as protection against ground moisture), it is necessary to ensure that water does not penetrate through the cracks into the building. In the place of projected joints in concrete, this is usually achieved by using special elements based on plastic, rubber, or bentonite strips. The problem is if the crack occurs out of the place of a projected joint. This happens very often (Figure 1). In such a case it is necessary to rely on the ordinary autogenous healing of concrete, but this works only under certain conditions, and depending mainly on the crack width. In the case of cracks wider than 0.1 mm, the ordinary autogenous healing of concrete is unreliable.

There some products exist to increase the autogenous healing ability of concrete. One of them is a crystalline admixture which causes secondary hydration in concrete. These additives are primarily intended to create a waterproof structure inside low-strength concrete. One side effect should also be a significant increase in the autogenous healing of concrete (according to the manufacturers’ claims), but the problem is that this assumption has never been proven by any independent tests.

The crystalline admixture is a dry material (powder) consisting mainly of Portland cement, and additionally containing fine quartz sand and other inorganic chemicals. The chemical recipes are subject to the protection of company patents in all manufacturers in the US, Canada, and the EU (Germany, Switzerland, Spain). The crystalline admixture’s waterproofing effect in concrete is achieved by the reaction of various chemical components contained in the solution when combined within the concrete matrix [1,2,3]. The process only works when the porous system of concrete reaches a sufficient level of moisture. The crucial part of the application is therefore perfect curing of the surface after the concreting is completed. The crystalline admixture in principle works by the chemical components chemically reacting with the cementitious matrix in the process of hydration with a temporary formation of Ca(OH)_2_, and a subsequent formation of disilicate and polysilicate anions. It is probable that the cumulative process is accompanied by the formation of 3Ca·2SiO_2_·3H_2_O, together with the creation of 3CaO·Al_2_O_3_·Ca(OH)_2_·12H_2_O [4]. The physical result of this chemical reaction is the creation of needle-shaped crystals inside the pore structure of the concrete. These crystals narrow the pores in the cement paste, and thus create a waterproofing structure. This modified concrete can stop the penetration of pressurized water, which has been proven several times in the past [5,6,7,8,9]. The waterproofing of concrete with crystalline admixture has also been proven by numerous laboratory studies [10,11,12,13,14,15,16,17], as well as the durability of this concrete [18,19,20,21]. Furthermore, the durability in aggressive environments was successfully tested as well [22,23,24,25].

The results of the above mentioned studies were very convincing; they confirm the waterproof and durability effect of crystalline admixture in concrete. Several studies have also confirmed the importance of curing of fresh concrete with crystalline admixture [26,27,28]. 

In addition to the aforementioned studies, there have been some other publications focused on the crystalline technology, which include problems with curing [29], fiber concrete issues while using crystalline admixtures [30] and for the combination of fly-ash concrete with crystalline admixtures [31], and the influence of early water exposure on a modified cementitious coating [32].

Likewise, the authors of this article have written several papers dealing with selected problems in the field of crystalline waterproofing systems: crystalline admixtures and their effect on selected properties of concrete [33], description of crystalline coating properties [34], speed of the crystalline admixture’s waterproofing effect in concrete [35], interaction between the crystalline coating and carbonated concrete, durability of concrete with a crystalline admixture in the earth environment [36], the effect of crystalline admixture on concrete and cement mortar compressive strength, and the long-term sorption properties of mortars modified by a crystallizing admixture [37,38,39].

The latest knowledge shows that organic compounds can also be theoretically used for improving self-healing ability, but with unreliable results. Some kinds of bacteria and fungi can, under very specific conditions, improve the self-healing ability of concrete, but this is difficult to use in practice [40].

### Motivation

Based on previous studies and experience from practice, it is obvious that crystalline admixtures improve the self-healing ability of concrete. However, this phenomenon has not yet been described by any independent study, and not at all in relation to textile reinforced concrete (TRC). Therefore, the aim of this study was to describe this process in detail, with a focus on the use in TRC. The topic of the research was inspired by a question from the British company, Mott MacDonald, addressed to a co-author (J. Pazderka) by Dr. Noushin Khosravi. The replacing of part of the conventional secondary structural reinforcement by crystalline admixture was the main question. The need to ensure an acceptable level of crack development in concrete while reducing the structural reinforcement led us to the idea of using TRC. A detailed description of the interaction between TRC and the crystalline admixture’s self-healing ability could create new uses for this composite. First, preliminary tests on cubes were created, after that specific boundary conditions (i.e., temperature, humidity, water exposure) applied to the TRC specimens were realized.

## 2. Materials and Methods

### 2.1. Preliminary Tests on Cubes

First, the basic parameters of the improved self-healing ability (due to crystalline admixture) were tested. Pure concrete was used for this initial phase of testing.

The experiment started in November 2015 and finished in June 2018. The aim of this research was to load these cracked samples with water pressure to determine their waterproofness, in terms of the crystalline admixture’s influence on the sealing of cracks. A water pressure test, using a modified methodology according to [41], was used.

All specimens were standard concrete test cubes 150 × 150 × 150 mm^3^ [42]. The specimens’ composition: Cement type CEM I 42.5, aggregate, and polypropylene fibers Forta Ferro (Grove, PA, USA) (Table 1). The length of the fibers was 54 mm, and the tensile strength was 620–758 MPa. The most commonly used product in practice worldwide, Xypex Admix (Richmond, BC, Canada), was chosen as a crystalline admixture for the study. It is known that in Xypex Admix admixture are only inorganic compounds; the specific composition is know-how of the producer.

Various methods of crack creation were considered. The creation of a realistic crack (if possible) in laboratory conditions is difficult, but it was necessary for the planned experiments. If the crack in the specimen does not resemble a real crack, then the experiment is valueless. Therefore, one of the objectives was to find a method for creating a realistic crack, with a predefined width. It was found (during the development of the methodology) that it was possible to make a realistic crack, but without a predefined width, or it was possible to make an unrealistic crack, with a predefined width. The first possibility was chosen. As a result, a method based on polypropylene fibers added into the concrete mixture was chosen for making realistic cracks in the test cubes. The cracks were made using a machine for testing the tensile splitting of concrete. Thanks to the presence of fibers, the cracks were created, but the concrete specimen was not destroyed. This method allows creating specimens with cracks, but the width of the cracks could not be controlled in advance. The crack width was measured by an Electrometer 900 optical microscope. Width of the cracks was in the range of 0.05–0.8 mm.

Concrete specimens were cured under polyethylene foil for 48 h after the concrete embedment and, consequently, the formwork was stripped, and they were placed in special tanks full of water to stop autogenous shrinkage [42]. The cracks were created 14 days after concreting.

An important side objective of the experiment was to qualify the environment for saving the specimens (in relation to the maximum efficiency of the secondary hydration process in cracks). The aim was to determine whether air is necessary to cause chemical reaction or not. The basic presupposition was that ideal conditions for the growth of crystals are a specific combination of moist and dry environment. At first, all cubes were held in tanks under the water level. Then, 45 days after embedment, the specimens were placed in tanks with a humid environment (Temperature: 20 °C; Humidity: 95%) (Figure 2a), without direct contact with liquid water. Humidity was controlled using a device based on the Arduino UNO platform. The cubes were held in this environment for the next 65 days.

Subsequently, the cubes were placed in boxes (Temperature 20 °C, Humidity: 60%), where the level of water was maintained at 20–30 mm from the bottom (Figure 2b). This caused capillary action through the cracks, and the subsequent evaporation of water to the environment. By this process, there was an attempt to simulate the environment of a real on-site structure.

### 2.2. Interaction between TRC and Improved Self-Healing Ability Caused by Crystalline Admixture

The main experiment was divided into three phases (Table 2). The first phase was focused on the influence of temperature and humidity on the self-healing ability of concrete. The second phase was focused on the influence of time and different conditions before the exposure of the test specimens to boundary conditions (curing of the specimens in a standard laboratory environment; temperature 20 °C, humidity 60%, and under water level). The last phase was focused on determining the influence of different cement types (CEM I and CEM II) on the self-healing capacity of concrete with crystalline admixtures. This was mainly done in order to classify the results of the first and the second phase (cast with CEM II) in comparison to the preliminary test (cast with CEM I). (Section 2.1).

The main aim of the experiment was to find out, if the crystalline admixture was able to help overgrow a large amount of the microcracks, whose formation is typical for thin structures made of TRC. This ability is important for the extension of the life of structures made from TRC.

The temperature and air humidity of the environment were measured using a thermometer and hygrometer based on the Arduino UNO platform. Measurements lasted for the whole time of the experiment duration.

The dimensions of the test specimens were 100 × 100 × 10 mm^3^ (the first phase) and 50 × 50 × 10 mm^3^ (the second phase and the third phase). The composition of all test specimens in all phases was the same (Table 3). For all test specimens in all phases, nonwoven polypropylene fabric, FILTEK (Jinan, China),, with a density of 200 g/m^2^ was used. The tensile strength lengthwise was 12 kN/m^2^, and transverse tensile strength was 7.5 kN/m^2^. There were two layers of polypropylene fabric in each test specimen, placed on both surfaces (distance from surface was 0.1 mm). The water–cement ratio was 0.4, this was caused by using fine aggregates and the polypropylene fabric.

In the first phase, 9 test specimens were produced, in the second phase, 56 test specimens were produced, and in the third phase, 6 test specimens were produced containing a crystalline admixture (Xypex). The same quantity of reference specimens was produced as well.

The test specimens were cured for two weeks, during which they were covered with a polyethylene foil to prevent the evaporation of moisture into the environment. During the curing, the temperature and humidity of the surrounding environment were monitored continuously by the Arduino UNO platform (Ivrea, Italy). Cracks were made by applying a bending load. The maximum crack width was about 0.1 mm, and the measurement of all crack widths was performed with a DigiMicro Profi II digital microscope (Leer, Germany), (Figure 3).

In the first phase, the test specimens were placed in an environment with different boundary conditions (Table 4).

The crack development was monitored for the next 4 weeks, with control imaging every 7 days using the DigiMicro Profi II microscope (Leer, Germany). For specimens placed in direct contact with water, evaporation was allowed to the surrounding environment with ordinary relative humidity (40–80%). For specimens placed in the environment with a 100% relative humidity, a decrease in humidity values can be observed at regular intervals. This reduction was caused by regular monitoring of the test specimens, which were removed from their environment for a short time for the purpose of photographic documentation.

The aim of the second phase was to determine the effect of curing on the activation of the crystalline admixture in concrete. 

Initial imaging was performed using the Digi Micro Profi II microscope at 50× magnification, and the specimens were placed in different environments based on a predetermined experimental scheme (Figure 4). The numbering of the test specimens with the Xypex admixture and the reference test specimens was always the same depending on the environment which was used. For each specimen, the same area was always captured by the microscope for the duration of the experiment.

The test specimens were divided into sets of four test specimens. The first two sets (test specimens 1–8) were placed immediately in the environment where the water level was maintained throughout the experiment so that the upper surface of the specimens was in contact with air (Figure 5). The remaining sets of test specimens were divided in half, the first half were fully submerged under the water level, and the second half were kept under laboratory conditions (temperature of around 25 °C, relative humidity of around 60%) without contact with liquid water. Subsequently, the two sets of test specimens were put in evaporation tanks; one in the water tank, and the other placed in laboratory conditions (temperature of around 25 °C, relative humidity of around 60%) without contact with water. The specimens were put in evaporation tanks at different time intervals: 7 days after the start of the experiment, after 14 days, 21 days, 28 days, 2 months, and 3 months. (Figure 6).

The last phase was focused on comparing the effectiveness of the crystalline admixture depending on the type of used cement. Two types of cement were tested, CEM I (Table 5) in the preliminary test, and CEM II in the first phase and the second phase. 

A total of 12 test specimens with dimensions of 50 × 50 × 10 mm^3^ were produced, i.e., a total of two sets of test specimens of three pieces each for each type of cement, and the corresponding number of reference specimens, were made.

## 3. Results and Discussion

### 3.1. Water Permeability and Material Analysis—Preliminary Tests

The water pressure test to determine the depth of water penetration was performed in three different time periods for cracked and uncracked specimens. The first test was performed 90 days after embedment, the second was performed 190 days after embedment, and the third was performed 945 days after embedment. (Table 6, Figure 7). For each test time period, three cracked specimens and three reference specimens (uncracked) were chosen.

The water pressure test for cracked test specimens was based on the [41] methodology. However, it was modified in terms of the applied water pressure level (0.2 MPa in our tests in comparison to 0.5 MPa from EN 12390-8). The lower level of water pressure was chosen with regard to the considered constructions; substructure of buildings. In these cases, the structure will never be exposed to a water pressure higher than 20 m of water column (0.2 MPa). For specimens that contained cracks 0.2 mm wide, a water flow occurred. In the case of cubes that contained cracks 0.1 mm wide or smaller, there was no leakage problem. From this perspective, it can be concluded that the crystalline admixture was able to heal cracks up to a width of 0.1 mm. This is reliable at a water pressure level of 0.2 MPa according to the results of the laboratory tests.

In all tested specimens, there was a crystal growth of white macroscopic crystals in the cracks. Material analysis (Figure 8) of the crystals was performed on the Spectro Xepos apparatus. The results showed that the crystals were mainly formed by calcium oxide CaO (85.28%). Silica (9.71%) and iron oxide (1.90%) were also present in the crystals (Figure 8).

### 3.2. Microscope Analysis—Results

The scientific software, Fáze (Version 5), developed by K. Forstová under the guidance of assoc. prof. J. Němeček at CTU in Prague, was used for the quantification of changes due to the crystal growth. The main advantage of the program is its ability to recognize pores and cracks in the concrete structure. In order to compare individual areas of interest, an initial measurement was taken as a reference, and the area of cracks was marked 100%; i.e., no cracks are healed. Any subsequent crystal growth then led to a reduction in the crack area, and thus to a percentage loss (Figure 9).

A Digi Micro Profi II microscope and a Canon PowerShot digital camera were used for the monitoring of the test specimens during the whole experiment in all phases (Table 7).

The measurement of the crack area during the first phase using the Fáze software (Version 5, Prague, Czech Republic) was performed at the beginning, before placement in individual experimental environments, and then at each test specimen inspection, i.e., 7, 14, 21, and 28 days after the start of the experiment. The values for each test specimen were then averaged and compared with each other (Table 8).

The averaged values of each test specimen were subsequently plotted in a graph for comparison (Figure 10 and Figure 11). The measured data showed a slight variance of values, which was caused by the quality of photographs used for analysis by the Fáze software; slight deviations in the order of percentage units were caused by the different moisture contents of the test specimens during the measurements, and changes in the test specimen illumination. 

For specimens that had not been in contact with water, there were no, or only minor, changes, both for Xypex and reference specimens. Crystal overgrowth could be observed in specimens that had been placed in contact with water. For specimens with the Xypex Admix additive, a decrease of approximately up to 33% in the original crack surface was measured. For the reference specimens placed in the water environment, this decrease was about 53%. The results of the measurements also showed that the temperature was almost irrelevant for the secondary hydration process (in the room temperature range), the most important parameter for crack healing was partial contact with water, allowing water to rise through the specimen, and its subsequent evaporation into the environment (Table 8).

The measured values gained during the second phase (Figure 12) were averaged for test specimens from the same group, and were plotted in summary graphs (Figure 13 and Figure 14). The groups were composed depending on the type of storage of the test specimens before storage in the evaporation tanks, i.e., the values for bodies marked 1–8 (stored in evaporation tanks after curing), 9–12 (one week below the water level), 13–16 (one week in a dry environment), 17–20 (14 days under water), 21–24 (14 days in a dry environment), 25–28 (21 days under water), 29–32 (21 days in a dry environment), 33–36 (28 days under water), 37–40 (28 days in a dry environment), 41–44 (2 months under water), 45–48 (2 months in a dry environment, 49–52 (3 weeks below the water surface), and 53–56 (3 weeks in a dry environment).

The obtained data confirmed the crystal growth in both sets of test specimens, with the crystalline admixture and reference specimens. In the reference test specimens, this overgrowth was smaller than in the specimens with the additive (Table 9).

In the case of the test specimens, which had been placed below the water surface before being placed in evaporating tanks, cracks were already overgrowing with crystals in this environment, and this continued even after their placement in evaporating tanks. For the specimens placed in evaporating tanks from the aquatic environment in the initial stages, the crystal growth in cracks was faster than for the specimens placed there in later stages, however, for the specimens placed in evaporating tanks in later stages, the resulting crystal growth in cracks was greater; it had largely already occurred in the aquatic environment.

For the test specimens placed in a standard environment (temperature around 25 °C, relative air humidity around 60%) before being placed in the evaporating tanks, the cracking showed a similar trend as for specimens 1–8, which were placed in evaporating tanks immediately after the end of curing.

The results of the third phase show that the growth of crystals in the cracks was not dependent on the used cement type. The resulting crack area for specimens with the Xypex additive was around 30% in both cases. The resulting size of the crack area of reference specimens was in both cases around 59% (Figure 15). As a result, it is possible to compare the data obtained in phases one to five.

### 3.3. Microstructure Analysis

The sealed cracks were studied using an optical microscope (Figure 16 and Figure 17). Images were taken of specimens from the series of the second phase, 134 days after the specimens’ creation. The images confirmed that the cracks 0.1 thick were healed by formations growing from the crack boundary surfaces.

## 4. Conclusions

The overall results of the performed tests showed that the interaction between TRC and the crystalline admixture’s self-healing ability brings benefits. One of the objectives of the study was to approximate the methodology of the experiment as close to practice as possible (influence of temperature, crack creation method, humidity, and type of used cement). For this reason, some of the experiments were conceived as long-term. Loaded TCR often exhibits a net of smaller microcracks, of which the dislocation is determined by the used fabric, and the difference with ordinary concrete. Whereas the self-healing ability of the studied admixture is limited with respect to crack depth, this application gives a suitable solution to reaching long-term entirety of the TRC element. The result is a combination of the advantages of a thin structure (allowed by TRC) and maintaining the waterproofness of the concrete (allowed by the crystalline admixture). One of the acquired findings of the study was, that this does not depend on temperature in the range of 4–30 °C (lower temperatures are of course problematic, as for most other cementitious materials). However, the interaction of both materials has its limits; the cracks must not be too wide (max. 0.1 mm), otherwise they will not heal. This was the main finding of the preliminary tests. However, the advantage is that it does not matter what type of cement is used (CEM I and CEM II showed the same results), and the composition of the newly formed crystals in the cracks corresponds to the composition of the C-S-H gel, so it can be assumed that the secondary hydration of Portland cement occurred in the crack area. This was confirmed by material analysis performed during the preliminary tests. Results acquired in the main test confirmed the initial assumption that the crystalline admixture improves the self-healing ability of TRC, as well as in the case of concrete with conventional reinforcement. Overgrowth of the cracks was more significant in the test specimens with crystalline admixture. The main output of the experimental program, which may be important for practical applications, is a detailed description of the kinetics of the self-treatment process under different conditions. Based on the presented results, it is possible to competently estimate the healing of cracks under specific conditions, which can be set artificially or may occur during construction. Artificial creation of conditions is much more complicated, but it compensates for disadvantages resulting from the presence of cracks in the structure.

## Figures and Tables

**Figure 1 materials-13-05787-f001:**
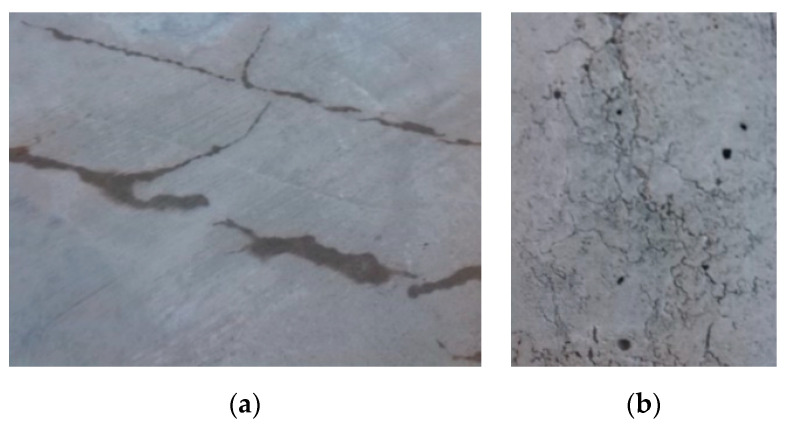
Cracks in concrete basement structures. (**a**) Water penetration through cracks in the ground floor. (**b**) Cracks in the wall in the building substructure.

**Figure 2 materials-13-05787-f002:**
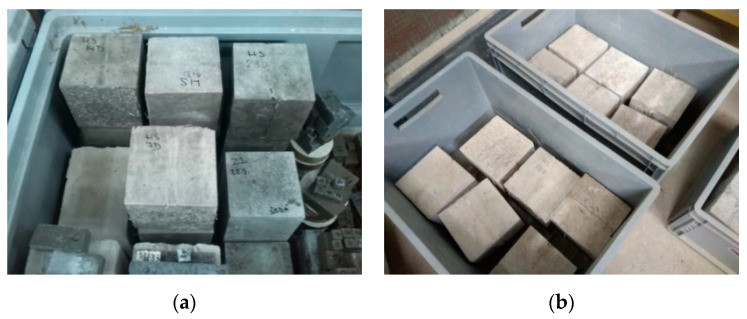
Placement of specimens. (**a**) Specimens in the tank with a humid environment. (**b**) Specimens in tanks with a water level about 20 mm above the bottom.

**Figure 3 materials-13-05787-f003:**
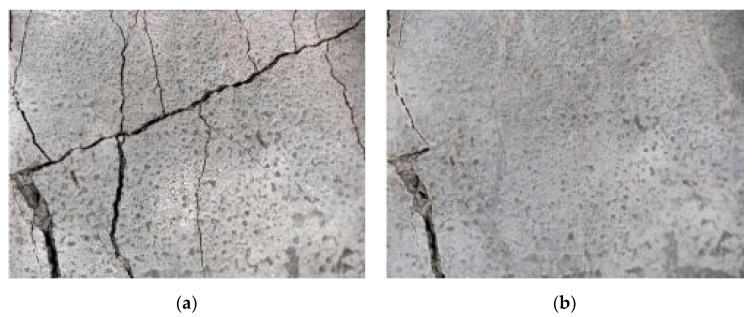
Output from the DigiMicro Profi II microscope. (**a**) First measurement. (**b**) Last measurement.

**Figure 4 materials-13-05787-f004:**
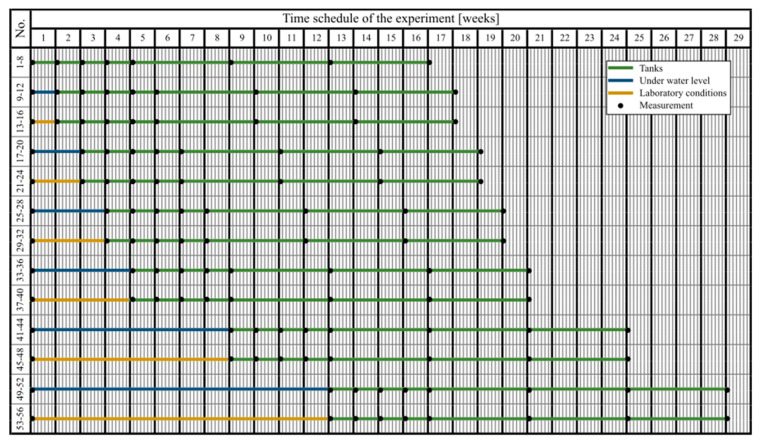
Time schedule of the second phase.

**Figure 5 materials-13-05787-f005:**
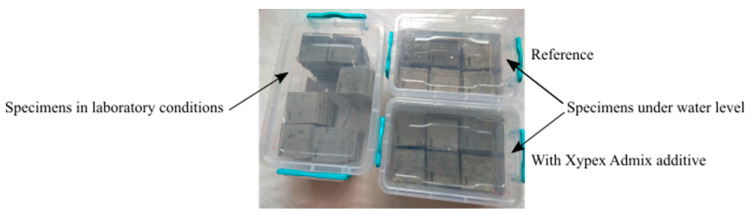
Storage of test specimens in the second phase.

**Figure 6 materials-13-05787-f006:**
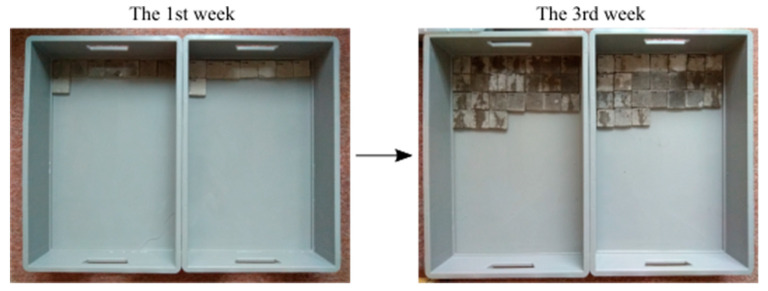
Specimens in tanks during the second phase.

**Figure 7 materials-13-05787-f007:**
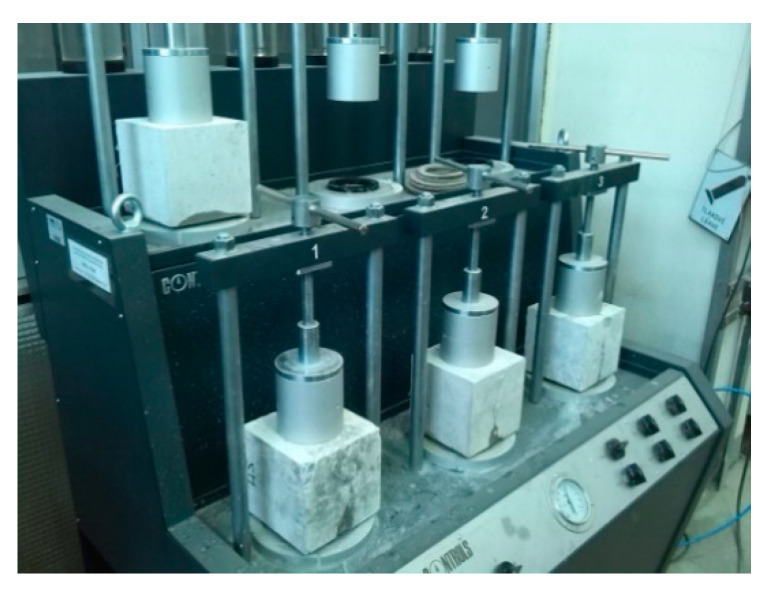
Testing of water penetration in cracked specimens.

**Figure 8 materials-13-05787-f008:**
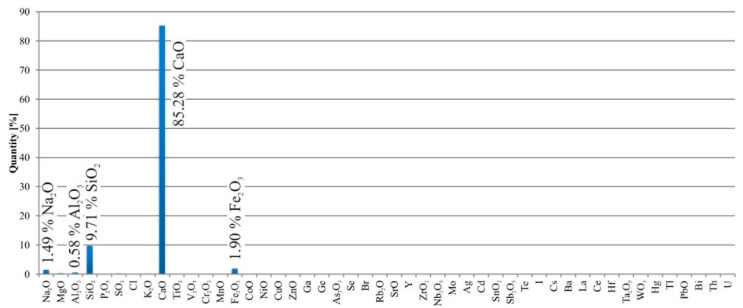
Material analysis of the crystals from the cube test specimens.

**Figure 9 materials-13-05787-f009:**
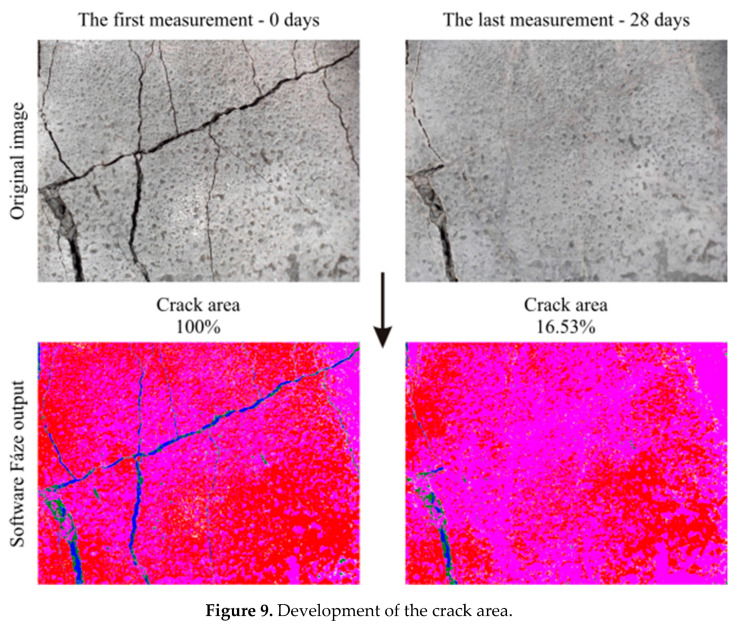
Development of the crack area.

**Figure 10 materials-13-05787-f010:**
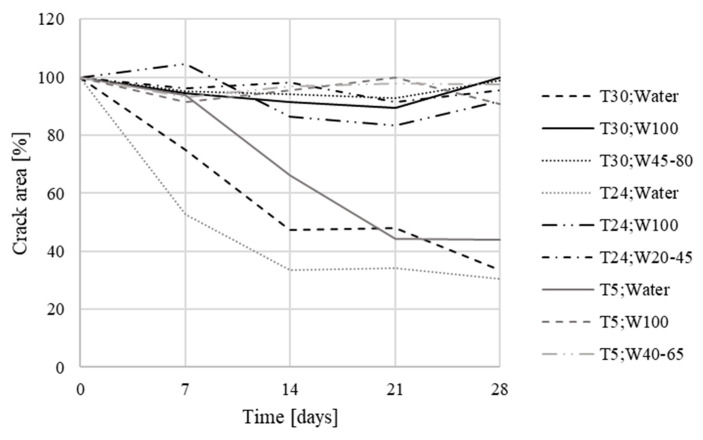
Crack development in specimens with Xypex Admix additive—the first phase (T = temperature, W = humidity).

**Figure 11 materials-13-05787-f011:**
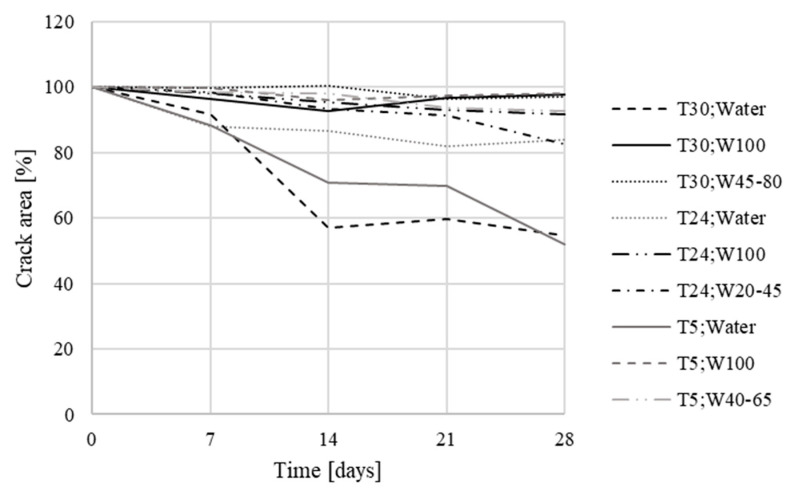
Crack development in reference specimens—the first phase (T = temperature, W = humidity).

**Figure 12 materials-13-05787-f012:**
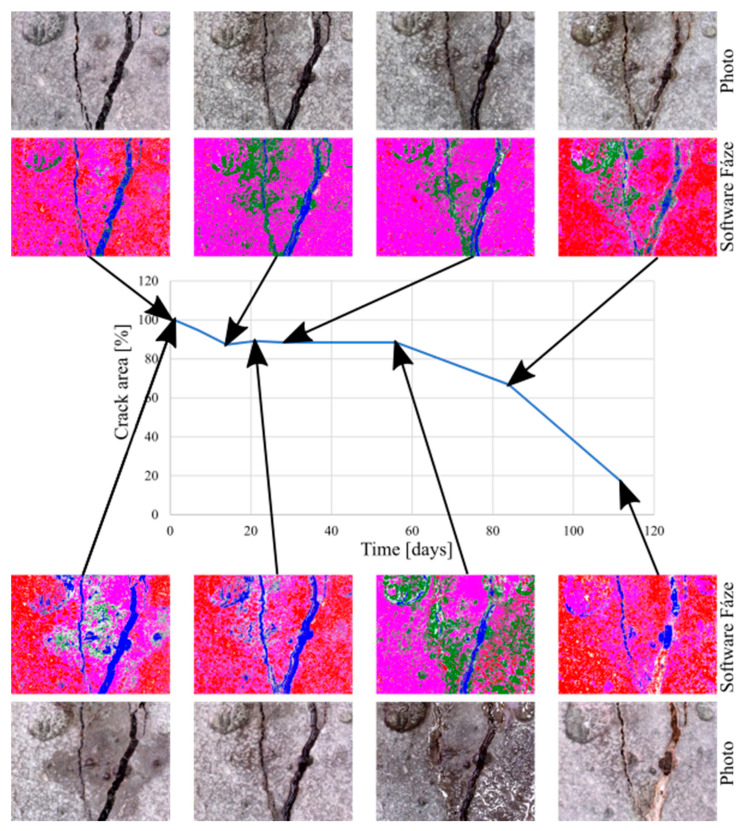
Microscope analysis and evaluation through Fáze software—graph creation.

**Figure 13 materials-13-05787-f013:**
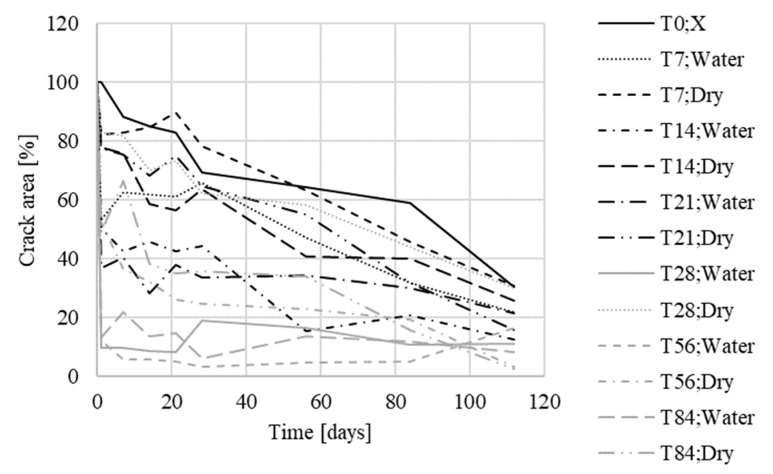
Crack development in specimens with Xypex Admix additive—the second phase (T = time of exposure).

**Figure 14 materials-13-05787-f014:**
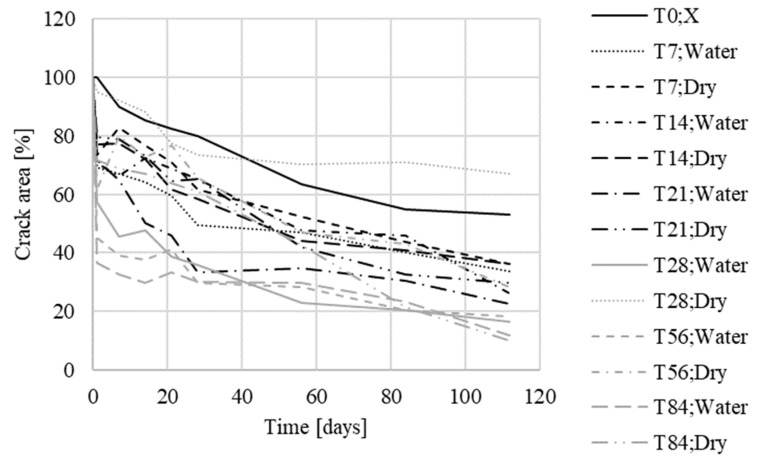
Crack development in reference specimens—the second phase (T = time of exposure).

**Figure 15 materials-13-05787-f015:**
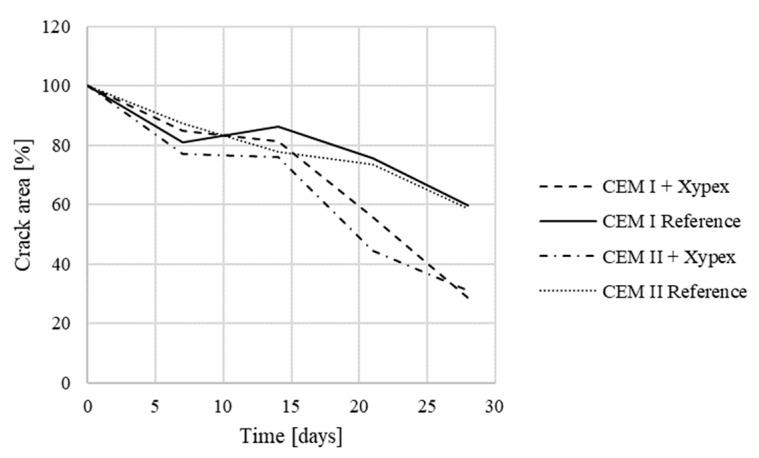
Crack development in specimens—the third phase.

**Figure 16 materials-13-05787-f016:**
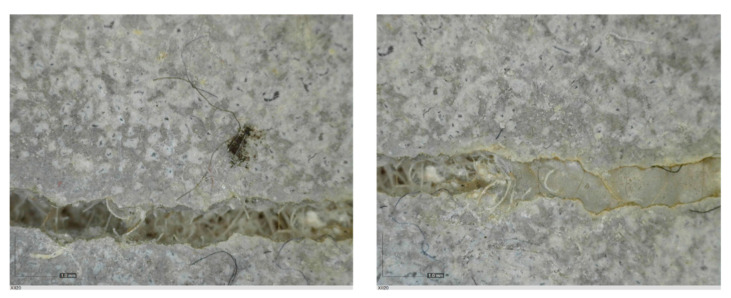
Cracks were healed by formations growing from the crack boundary surfaces (50× zoom).

**Figure 17 materials-13-05787-f017:**
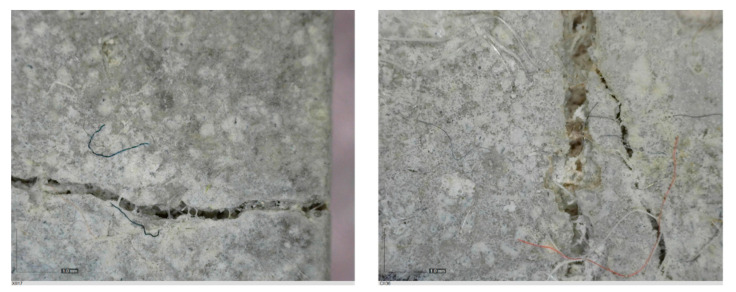
Cracks were healed by formations growing from the crack boundary surfaces (50× zoom).

**Table 1 materials-13-05787-t001:** Composition of specimens.

Composition	Weight (kg)
CEM I 42.5	23.100
Aggregate 0–4 mm	60.100
Aggregate 4–8 mm	35.700
Aggregate 8–16 mm	31.900
Xypex Admix C1000	0.347
Water	13.700
Forta Ferro Fibers	0.090

**Table 2 materials-13-05787-t002:** Testing program.

	Preliminary Tests	The First Phase	The Second Phase	The Third Phase
Time period	1/11/2015–25/6/2018	5/1/2018–16/2/2018	21/11/2019–4/6/2020	15/5/2020–26/6/2020
Humidity	55%	20–100%	60%	60%
Temperature	20 °C	5; 24; 30 °C	25 °C	25 °C
Determined parameters	Basic parameters of self-healing parameters	Different temp. and humid.	Different exposure time	Different cement type

**Table 3 materials-13-05787-t003:** Composition of specimens used in all phases.

Composition	Weight (g)
Cement CEM II	500
Aggregate 0–0.4 mm	500
Xypex Admix C1000	20
Water	200
Stachement plasticizer	2

**Table 4 materials-13-05787-t004:** Boundary conditions in the second phase.

Bound. Cond. No.	T (°C)	Relative Humidity (%)
1	30	45–80
2	30	100
3	30	direct contact with water on one surface
4	24	20–45
5	24	100
6	24	direct contact with water on one surface
7	5	40–65
8	5	100
9	5	direct contact with water on one surface

**Table 5 materials-13-05787-t005:** Composition of specimens made in the third phase.

Composition	Weight (g)
Cement CEM I	500
Aggregate 0–0.4	500
Xypex Admix C1000	20
Water	200
Stachement Plasticizer	2

**Table 6 materials-13-05787-t006:** Height of water penetration in specimens, crack width is given for each cracked specimen.

Spec. No.	90 Days			190 Days			945 Days		
	Uncr.	Cracked		Uncr.	Cracked		Uncr.	Cracked	
1	40.0	-	(0.6 mm)	62.0	73.0	(0.06 mm)	43.0	78.0	(0.03 mm)
2	79.0	68.0	(0.08 mm)	50.0	67.0	(0.08 mm)	39.0	64.0	(0.09 mm)
3	31.0	-	(0.35 mm)	54.0	-	(0.45 mm)	56.0	-	(0.33 mm)

**Table 7 materials-13-05787-t007:** Crack development of one test specimen during the second phase.

	After Finishing of Curing	0 Days	7 Days
General view	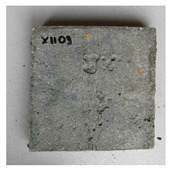	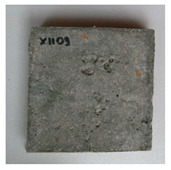	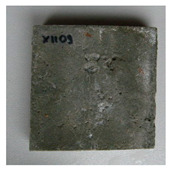
Microscope	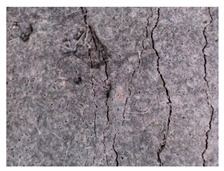	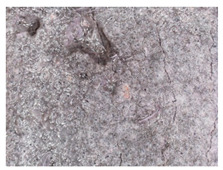	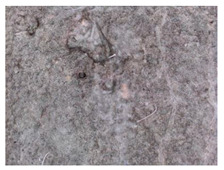
	14 days	21 days	28 days
General view	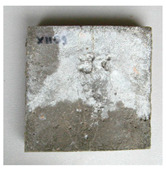	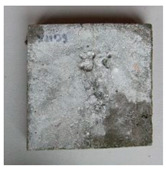	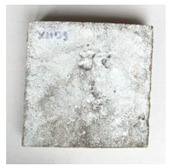
Microscope	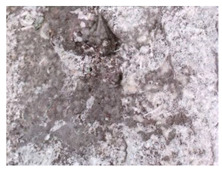	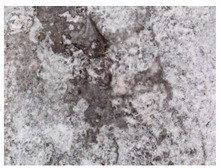	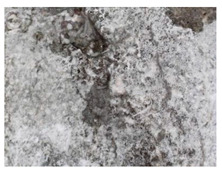
	2 months	3 months	4 months
General view	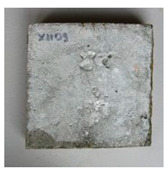	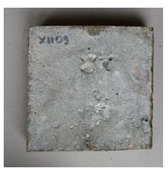	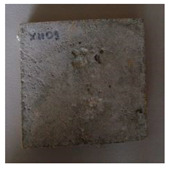
Microscope	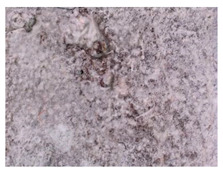	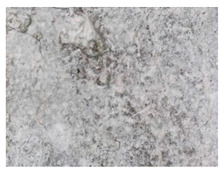	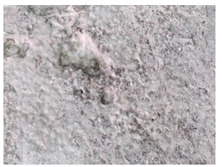

**Table 8 materials-13-05787-t008:** Comparison of crack development—the first phase.

Type	No.	Environment		Average Crack Development (%)				
		Temperature (°C)	Humidity	0 Days	7 Days	14 Days	21 Days	28 Days
Specimens with Xypex Admix	XII1	30	water	100.00	75.01	47.20	47.99	33.52
	XII2	24	water	100.00	52.80	33.63	34.17	30.39
	XII3	5	40–65%	100.00	93.21	96.99	97.82	97.62
	XII4	24	100	100.00	104.65	86.49	83.41	91.79
	XII5	30	100	100.00	94.54	91.47	89.58	99.94
	XII6	24	20–45%	100.00	96.04	98.03	91.52	95.38
	XII7	30	45–80%	100.00	95.17	94.24	92.83	98.99
	XII8	5	100%	100.00	91.30	95.41	99.81	90.65
	XII9	5	water	100.00	93.80	66.30	44.35	43.86
Reference specimens	CII1	5	water	100.00	88.43	70.77	69.72	51.83
	CII2	30	water	100.00	91.71	56.85	59.80	54.51
	CII3	24	100%	100.00	98.00	95.27	92.85	91.78
	CII4	30	45–80%	100.00	99.62	100.27	96.41	97.11
	CII5	24	water	100.00	88.01	86.61	81.89	83.86
	CII6	30	100%	100.00	96.43	92.61	96.74	97.72
	CII7	5	100%	100.00	99.58	96.11	97.23	98.05
	CII8	24	20–45%	100.00	98.31	93.21	91.40	82.70
	CII9	5	40–65%	100.00	98.29	98.16	93.60	92.62

**Table 9 materials-13-05787-t009:** Comparison of crack development—the second phase.

		Development of the Crack Area (%)								
	Type	Start	Env.	7 Days	14 Days	21 Days	28 Days	56 Days	84 Days	112 Days
Specimens with Xypex additive	1–8	100.00	100.00	88.22	84.99	82.84	69.42	64.20	58.89	30.19
	9–12	100.00	53.16	62.48	61.74	61.09	65.73	47.22	31.61	21.69
	13–16	100.00	82.33	82.97	84.80	89.61	78.40	63.20	45.62	30.72
	17–20	100.00	50.49	42.30	45.60	42.51	44.27	15.15	20.53	12.39
	21–24	100.00	77.84	75.59	58.43	56.33	63.41	40.59	39.97	25.63
	25–28	100.00	36.61	40.28	28.00	37.94	33.65	34.27	29.75	21.48
	29–30	100.00	77.62	75.32	68.11	75.13	64.69	55.09	31.86	15.64
	33–36	100.00	9.57	9.41	8.60	8.09	18.99	16.51	10.57	10.93
	37–40	100.00	83.01	81.79	69.88	73.15	62.15	58.11	43.72	30.09
	41–44	100.00	11.88	5.62	5.46	4.76	2.92	4.66	4.76	16.17
	45–48	100.00	52.39	36.43	31.90	26.12	24.58	22.65	19.05	3.26
	49–52	100.00	13.17	21.60	13.48	14.51	5.85	13.56	11.75	8.08
	53–56	100.00	49.06	66.27	38.27	34.77	35.67	33.82	15.55	2.54
Reference test specimens	1–8	100.00	100.00	90.03	85.14	82.47	79.91	63.60	54.68	53.11
	9–12	100.00	69.22	67.18	64.08	59.47	49.42	46.99	40.06	33.82
	13–16	100.00	73.57	82.69	76.72	71.08	61.63	52.40	43.83	36.23
	17–20	100.00	70.71	66.37	72.22	68.93	65.04	47.58	45.95	26.29
	21–24	100.00	77.15	77.52	72.12	61.71	58.25	44.09	40.75	36.40
	25–28	100.00	71.72	64.96	50.18	45.81	33.30	34.85	30.34	22.48
	29–30	100.00	79.43	78.82	72.91	64.67	65.36	41.85	32.67	29.89
	33–36	100.00	57.38	45.61	47.74	38.63	35.70	23.05	20.33	16.41
	37–40	100.00	94.77	91.98	88.21	77.21	73.58	70.28	70.88	66.90
	41–44	100.00	45.24	39.25	37.57	41.11	29.82	28.46	20.55	18.22
	45–48	100.00	61.65	79.74	72.62	76.78	65.60	47.07	43.12	28.46
	49–52	100.00	36.69	32.53	29.94	33.30	30.24	29.63	23.40	12.08
	53–56	100.00	71.63	68.73	66.90	63.85	60.91	42.56	21.56	10.24

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
