# Peer review of "Textile Reinforced Concrete in Combination with Improved Self-Healing Ability Caused by Crystalline Admixture"

_materials, 2020, doi:10.3390/ma13245787_

Round 1

Reviewer 1 Report

The paper deals with the self-healing ability of concrete with crystalline admixtures. Therefore, samples of thin-walled TRC elements were tested under different environments. The topic is interesting and crystalline admixtures provide opportunities for enhancing the impermeability of concrete against water. Unfortunately, the paper suffers from many shortcomings, which in my point of view do not permit publication in the current form. In the following a list with the most crucial shortcomings is given:

  • The title and motivation is misleading. What exactly has TRC to do with the performed research? Does the presence of textile reinforcement anyhow influences the self-healing capacity of concrete with crystalline admixtures? What exactly is the interaction between TRC and improved self-healing ability caused by crystalline admixture as mentioned in the title of chapter 3.2? Is the aim of the study to investigate the self-healing capacity of thin-walled plates (which may be reinforced with textile-reinforcement in order to avoid corrosion)? How was the thickness of 10 mm chosen? The paper opens up so many questions which are left unanswered due to the unclear description.
  • In the motivation (chapter 2) it is mentioned that the self-healing ability of concrete with crystalline admixture has not yet been described in any independent study. However, a search in the internet with some keywords (it took me about 20 seconds) revealed studies to exactly that topic; see for example:

[1] M. Roig-Flores, S. Moscato, P. Serna, L. Ferrara. Self-healing capability of concrete with crystalline admixtures in different environments. Construction and Building Materials, Volume 86, 2015, Pages 1-11, https://doi.org/10.1016/j.conbuildmat.2015.03.091.

[2] K. Sisomphon, O. Copuroglu, E.A.B. Koenders. Self-healing of surface cracks in mortars with expansive additive and crystalline additive. Cement and Concrete Composites, Volume 34, Issue 4, 2012, Pages 566-574, https://doi.org/10.1016/j.cemconcomp.2012.01.005.

  • It is not a good practice to make so many self-citations (eleven in the last paragraph of chapter 1) It would be better to just list two or three of the most valuable ones in order to give an overview of own research activities.
  • Chapter 2, line 91-93. Why to mention the names who came up with the idea when the actual question is not explained.
  • Chapter 3: The description of the test programme is relatively unclear. It may be significantly improved by:
    • An overview of the whole testing programme. This can be for example a table which includes the preliminary tests as well as the tests on the thin walled plates, including the boundary conditions and the aim of the test phase.
    • At which time after casting was the crack created?
    • Explanations are very vague, see for example Section 3.2: “The first phase was focused on the influence of temperature and humidity on boundary conditions” Which boundary conditions? Please specify.
    • Description of the fabric on page 6, line 161-163. Can you provide more information on the fabric? E.g. Mechanical properties? How and where was it placed inside the concrete?
    • A lot of pictures are given in this Section with no added value. E.g. Figure 5 and 6 are placed but not explained in the text. What is the added value of Figures 8-10 but also Fig. 3,4, 7 and 14.
    • The whole section could be much more comprehensive in order to draw a clear picture on what was the testing programme and its aims.
  • Chapter 4 also suffers from an unclear and unsufficient decription. The following list represents some aspects that are particularly disturbing:
    • Water permeability and material analysis: How was the permeability quantified? A list of test specimens with measured crack width and measured water flow would be really helpful. What crack-widths have been measured on their self-healing ability? No information is given at all. Just to state that with a crack width of 0.1 mm no water flow occurred is not sufficient in my opinion.
    • Was the water pressure also measured in the different time steps for the cracked specimens? If yes, what was the result? Again no information is given.
    • What is the added value of Fig 17 and 18?
    • Please reconsider the naming of the specimens in Section 4.2. An identification in the diagrams is almost not possible making Fig 21-22 and Fig 24-26 relatively worthless. An idea would be to name the specimens according to the boundary conditions. E.g. specimen No XII1 would be S T30 W 0 (S…Specimen T30…..Temperature W….Humidity-Water 0….days). This would allow an easy identification in the diagrams.
    • Also consider that the paper may be printed in black and white colors only. All the lines in Fig 21-22 and Fig 24-26 would look the same, making an identification again not possible. A combination of e.g. dashed or dotted lines with different greyscales would help.
    • Is there a difference between CII1 and CII2 to CII5 except of the temperature? The crack development differs about 20%. Is such a high scatter usual or was there a different problem? An effect of temperature seems to negligible according to the other test results.
  • Conclusions
    • The conclusions are a repetition of the abstract. The main results of the study should be highlighted in bullet points, in order to draw a clear picture of the outcomings.
    • Clarify what are new findings in context of the research in comparison to the current state of knowledge. E.g. the self-healing capability of concrete with crystalline admixtures in different environments has been investeigated before by other researchers. See literature source above.
    • Why is there a larger elastic limit in TRC? In my opinion there is state I (uncracked concrete) and if the concrete tensile strength is reached cracks occur. There is not that much difference to ordinary RC concrete. However, main difference can be the smaller crack distances, which may lead to smaller crack widths (in dependency of the used textile fabric).

Author Response

Dear reviewer,
Thank you for your valuable comments. For response please see the attachment.

Reviewer 2 Report

The paper presents an examination of the concrete with textile fibres regarding their cracks healing performance in connection with the special crystalline admixture. The topic is of interest, the research is well designed, the authors present interesting experimental results. However, the main shortcoming of the paper is the presentation style and the conception of the manuscript. The structure of the paper should be modified, e.g. some of the results are presented in methodological part. In addition, I have also these particular comment and recommendations:
1. Introduction - it is necessary to adjust the chemical formulas, which are presented in lines 57 and 58. Please add dots between the individual components (oxides, hydroxide, water).
2. Line 104 – please add the unit (mm) to the particular aggregate fractions – 0-4 mm; 4-8 mm; etc. The same applies to Table 1.
3. Please specify in Methods in more detail the fibres used by concrete production (size, length, any other characteristics and parameters relevant). In Table 1, you mentioned Forta-Ferro Fibres. Is that the fibre producer brand? Are these the polypropylene fibres as mentioned in the text?
4. The crystalline admixture should be also a little described. I understand that it is not possible to write the exact composition of the ingredient, but at least a brief description of the composition, or type of individual ingredients would be appropriate. In my opinion, it is not enough just to state that there are included chemicals.
5. More information are needed also in connection to the concrete curing process and the cracks formation. In the present form, the procedure is not clear enough. For instance, what time the cracks were performed? After 48 hours of curing? Which were the curing conditions? Tap water, under air, saturated lime solution? How the initial cracks were documented?
6. Please specify what you mean by humid environment – line 138. How the humidity was controlled?
7. Line 161 - Nonwoven Polypropylene Fabric mentioned in the text – this is the same material as given in Table 1 (Forta-Ferro Fibers)?
8. In my opinion, the Figures 5 and 6 should be presented in Results or if only demonstrated the experimental conditions, they should be omitted altogether.
9. Lines 188-198 - it would be better to present this information in a table so that the same text is not repeated.
10. Table 3, Figures 15 and 16 should be presented in Results.
11. In Results, at least a little discussion on the observations and findings and a comparison with the other authors is needed.
I propose a major revision to significantly modify the structure of the paper.

Author Response

(The authors gave the same response as above.)

Reviewer 3 Report

This manuscript investigates use of self-healing admixture on the properties of TRC. This contains practical information which is very helpful for readers who are not familiar with the subject. I only have minor comments as follows.

  • TRC in the title should be written as its full name
  • Most of figures (especially figs. 1-18) are not needed in the text. They should be completely removed or appear in Appendix. Fig. 19 should be tabulated, not appear as a figure.
  • In the introduction, other self-healing agents should be reviewed (i.e., bacteria)

Author Response

(The authors gave the same response as above.)

Reviewer 4 Report

The paper matches the Journal's items dealing with the problem of crack permeability and especially focusing on the relation between textile reinforced concrete (TRC) and the crystalline admixture’s self-healing ability.

The analysis of this interaction distinguishes the work from the other studies proceeding by investigating its advantages for an improved healing of concrete and the uses for this composite.

The paper sufficiently describes the aim and the methodology used for the performance of a water pressure test loading the cracked samples to determine the crystalline admixture’s influence on the sealing of cracks.

The experiment, with its three phases, the different boundary conditions and the sets of 4 test specimens, is illustrated enough in detail, also through figures.

Considering the temperature factor, the measurements of the crack area and the type of cement used, the test results are sufficiently discussed. The usefulness of the interaction investigated in the process of formation of microcracks through the advantages of a thin structure and the crystalline admixture’s waterproofing effect in concrete is finally showed.

The study is interesting but the paper as presented in this version is not satisfactory in some aspects. The authors are thus recommended to follow some suggestions for the amelioration of their work to be published.

- The abstract and the conclusion section are formulated in the same way. Please modify them trying not to use the same words but writing them differently.

- The motivation section could be inserted in the previous section, forming a unique one with the Introduction.

- For a clear and easier reading of the paper, the authors are invited to include at the end of the Introduction section a brief scheme of the content of the following sections in which the paper is structured.

- Some sentences are often repeated entirely with the same words in some sections of the paper (i.e. Section 3 and Conclusions). Please express the concepts with different words.

- As far as the English language is concerned, in general it is sufficiently fluent even though some sentences could be changed in the paper. There are not many particular errors to be corrected, only few sentences should be checked for corrections.

Section 3

- The experiment lasted for 2.5 years due to the evolution of material during the secondary hydration process.

- The specimens’ composition: Cement type CEM I 42.5, aggregate with a fraction of 0-4, 4-8 and 8-16, polypropylene fibers (Table 1).

- The question was whether the crack overgrowth process could work under water, or if the access of air is necessary for the right chemical reaction.

- The temperature and air humidity of the environment was measured using the thermometer and hygrometer based on the Arduino UNO platform for the whole time of the experiment duration (Figure 5, Figure 6).

- The objective of the second Phase was to determine the influence the exposure time to water from the end of curing on the activation of the secondary hydration process caused by the crystalline admixture in concrete.

- The numbering of the test specimens with the Xypex admixture and the reference test specimens was always the same depending on the environment which was used, for simplicity see the following schedule (Figure 12).

Section 4

- For the test specimens which had been placed in a standard environment

- It is appropriate to put all figure legends below each figures in a centered position (i. e. Check figures 2, 3 and 7). Please correct all the figures.

Author Response

(The authors gave the same response as above.)

Round 2

Reviewer 1 Report

The manuscript has been improved. However, some issues remain, which should be carefully considered before final submission

  • The whole manuscript should be checked for bad wording and misspelling, e.g:
    • Abstract line 18-20: The result is a combination of the advantages of a thin structure (allowed by TRC) and maintaining the waterproofness of concrete. Bad wording; Suggestion: This allows for the creation of advantageous thin (achieved by TRC) and waterproof (achieved by the crystalline admixtures) concrete structures.
    • Page 4, line 130-131: The question was if the access of the air is necessary to the chemical reaction. Bad wording; Suggestion: The aim was to determine whether air is necessary to cause chemical reaction or not.
    • Page 4, line 148-149: The last phase was focused on the differences between cement CEM I and CEM II, because of the comparison with the cubic test specimens made during the preliminary experiment. Bad wording; Suggestion: The last phase was focused on determining the influence of different cement types (CEM I and CEM II) on the self-healing capacity of concrete with crystalline admixtures. This was mainly done in order to classify the results of Phase I and II (casted with CEM I) in comparison to the preliminary test (casted with CEM II).
    • Page 6, line 196-197: … to determine the influence the exposure …. Bad wording
    • Page 8, line 241-242: The water pressure test for cracked test specimens was based on EN 12390-8 methodology and it was modified (applied water pressure level 0.2 MPa). Bad wording; Suggestion: The water pressure test for cracked test specimens was based on EN 12390-8 methodology. However, it was modified in terms of the applied water pressure level (0.2 MPa in the own tests in comparison to XX from EN 12390-8). Give a short explanation why you have modified the water pressure level.
  • Be more specific, e.g:
    • Page 4, line 148: The second phase was focused on the influence of time and different conditions before the exposure of the test specimens to boundary conditions; Again: What boundary conditions?
    • Page 4, line 153: This ability is important for life extension of structures made from TRC; Why? I thought the paper deals about providing a waterproof structure and not about life time extension. How can a life time extension be achieved by sealing the micro cracks as there is no risk of rusting in case of textile reinforcement made of non-metallic materials?
    • The composition of concrete in the second and third phase (table 3 and 4) is given. What about the first phase? I guess it was the same as in table 3. If so please correct table header
    • Page 5, line 163: What was the distance from the surface?
  • In the whole manuscript there are many line breaks that are not useful at all. The manuscript would be more comprehensive and clearer without them, e.g:
    • Page 2 line 65 to 79 or page 3, line 97-104
    • Section 2.2 and 3.2: Why not paragraphing every test phase (without any line breaks within). Then it would be clear at first sight what the first phase, the second and third phase is.
  • Section 2.1: give reference to table 2.
  • Table 2, Suggestions:
    • Replace “Aim of the phase” with “Determined parameters”.
    • Write “ basic parameters of self-healing capacity.
  • Page 5, line 175:
    • After that, cracks were made using the bending moment. Bad wording; Suggestion: Cracks were made by applying a bending load.
    • Please discuss: Is there an influence of the type of crack, as the cracks in the preliminary test were initiated by a splitting device and the cracks in the TRC plates were initiated by applying a bending load? In the latter the cracks are getting smaller the close one gets to the concrete compression zone.
  • Table 4, Suggestions:
    • Consider using abbreviations in Table 4, as the rows consume a lot of unnecessary space (e.g. BC instead of Boundary conditions).
    • Make the table header more precise: E.g. Boundary conditions in Phase I.
  • Please give also values of water penetration in Table 6 for cracked specimens; additionally include crack widths.
  • Page 11, line 273. Add information that this paragraph is about Phase I.
  • Figure 10, 11 and 13,14
    • Give short explanation on the legend (e.g. in the figure description)
    • Why not using the same color and style in Fig 10 and Fig 11 for the same type of boundary conditions. E.g. T24, W100 is continuous line in Fig 10, while it is a bar-dotted line in Fig. 11. This makes it more difficult to compare the two figures. If the color and style would be the same one could easily draw conclusions at first sight. Same can be stated for Fig 13 and 14.
  • Fig 13: Please discuss why there is a decreasing crack area in the beginning also for dry conditions while in Phase I a significantly decreasing crack area was only detected in the case of water storage
  • Fig 15: Please sort legend accordingly
  • Conclusions:
    • Please discuss the influence of environment in the conclusions and the conditions in reality. Are the results of the experiments transferable to reality?

Author Response

Dear reviewer,
Thank you for your additional comments. For response please see the attachment.

Reviewer 2 Report

The manuscript was revised according to the reviewers´recommendations.

I have no more comments.

Author Response

Thank you for your approval and your valuable comments.